# Use of Complementary and Alternative Medicine by Greek Patients with Inflammatory Bowel Disease

**DOI:** 10.3390/nu16213679

**Published:** 2024-10-29

**Authors:** John K. Triantafillidis, Aristofanis Gikas, Georgia Kontrarou, Manousos Konstantoulakis, Apostolos Papalois

**Affiliations:** 1Metropolitan General Hospital, 15562 Holargos, Greece; jktrian@gmail.com (J.K.T.); gkontrarou@yahoo.gr (G.K.); 2Health Center “Kalyvia”, 19010 Athens, Greece; argikas@ath.forthnet.gr; 32nd Department of Surgery, School of Medicine, University of Athens, Aretaieion University Hospital, Vas. Sophias 76 Av., 11528 Athens, Greece; mkonstad@med.uoa.gr

**Keywords:** ulcerative colitis, Crohn’s disease, complementary medicine, alternative medicine

## Abstract

Background and Objectives: Complementary and alternative medicine (CAM) is regularly used by several patients with inflammatory bowel disease (IBD) in many countries. Data concerning the use of CAM in Greek patients with IBD are lacking. This study aimed to determine the prevalence and indicators of CAM use in Greek IBD patients. Patients and Methods: Adult patients with IBD attending our specialized IBD department in “Metropolitan General” Hospital, Holargos, Greece, completed a special questionnaire regarding the use of CAM. Several clinical and epidemiological characteristics were recorded. The results were compared with a disease-control group (patients with irritable bowel syndrome or gastroesophageal reflux disease). The study outcome was the prevalence of CAM use in both groups. In this study, 270 patients, of whom 128 were female, with IBD (Crohn’s disease 134 and ulcerative colitis 136) and a median age of 42.3 ± 17.5 years (range 31–52), and 138 patients serving as the disease control group were analyzed. Results: The prevalence of previous and current CAM use in patients with IBD was 36.3% versus 27.5% in the control group (no significant differences). No significant differences were also noticed between the patients with either CD or UC. In the multivariable analysis, university education and treatment with steroids and TNF-α inhibitors were independent social indicators of CAM use. Conclusions: The percentage of CAM use by the Greek patients with IBD was quite high, similar to other European countries. Although numerically higher, this percentage was not significantly different compared with the disease control group. The use of CAM in IBD patients in Greece is associated with a higher educational level, and treatment with steroids and TNF-α inhibitors.

## 1. Introduction

Inflammatory bowel disease (IBD), i.e., Crohn’s disease (CD) and ulcerative colitis (UC), is the most important chronic inflammatory enteropathy in humans [1]. They are chronic, recurring, and incurable diseases that require complex laboratory investigation to reach the correct diagnosis and to assess their activity, as well as a combination of drugs to treat them efficiently. An estimated 1.4 million people in the U.S. and 2.2 million people in Europe suffer from the disease. Although the etiology of IBD is largely unknown, it is considered to be multifactorial with the participation of environmental, genetic, and immunological factors, with factors related to the intestinal flora being the most important. Ultimately, mucosal inflammation develops in predisposed individuals due to an excessive mucosal immune response directed against antigens of the intestinal lumen. Treatment consists of oral or parenteral administration of various anti-inflammatory, immunosuppressive, and immunomodulatory agents [1].

Complementary and alternative therapies (CAM) are defined as therapies that are presently not considered part of conventional medical practice. They are termed “complementary” when used in addition to traditional therapies and “alternative” when used instead of conventional therapies. There are different types of CAM that could originate from (i) medical systems (homeopathy, traditional Chinese medicine, and Ayurveda); (ii) mind–body medicine (meditation, prayer, and healing); (iii) natural products (herbs, dietary supplements, vitamins, minerals, and probiotics); (iv) psycho-somatic techniques and physical practices (chiropractic and massage); and (v) energy medicine (Qigong, Reiki, therapeutic touch, and the use of magnetic fields) [2]. According to the U.S. National Center for Complementary and Integrative Health (NCCIH), CAM therapies include a wide spectrum of practices and products, either biological (e.g., herbs or botanicals, vitamins, minerals, probiotics, homeopathic products, and Chinese herbal remedies) or non-biological (e.g., prayer, meditation, music therapy, and yoga) [3].

The prevalence of CAM use in IBD patients in different countries is relatively high, ranging between 21% and 60% [3,4,5,6,7,8]. Its use in IBD patients is constantly increasing as they look for ways to treat the disease beyond conventional treatment. Reasons for using CAM include patients’ desire for a holistic approach to disease, their belief that herbal medicines are natural and non-toxic, a lack of response or unwanted effects from conventional treatment, and their desire for better disease control. In addition, the use of CAM is favored by psychiatric co-morbidities, difficulties in the patient–physician relationships, long disease duration, long-term use of corticosteroids, and other factors, including high annual income, educational, and economic status. However, despite the widespread use of CAM, 75% of IBD patients do not discuss the use of CAM with their doctors, although the available data suggest that some of these therapies could enhance the action of conventional drugs. The cost of treating IBD is now becoming prohibitive in developing countries, especially those with huge populations. CAM could improve the economic parameters of treating IBD in the long term. Nevertheless, the hesitancy in their application by gastroenterologists is evident for essential reasons, e.g., the absence of extensive multicenter randomized clinical studies and the lack of knowledge of the scientific data concerning CAM on the part of the therapists.

Thus far, there are no data in Greece regarding the use of CAM by patients with IBD. This study aimed to record the epidemiological parameters and the type of alternative forms of treatment of IBD adopted by Greek patients in comparison with a disease control group.

## 2. Patients—Methods

A detailed questionnaire was used regarding alternative therapy by patients with UC and CD. The questionnaire included 30 questions about the type and frequency of CAM use. We recorded queries related to the use of plants and herbs (e.g., mastic gum and aloe), lifestyle modifications (e.g., exercise, diet, and smoking cessation), as well as psychosomatic therapies—i.e., massage, kinesiology, homeopathy, and acupuncture—and data concerning demographic parameters such as age, gender, education, marital status, employment status, diet, and lifestyle habits. The questionnaire also included disease characteristics, the type of IBD, current activity, duration of disease, and type of conventional medications. One of the authors (JKT) completed the questionnaire in all patients and the control group following a detailed personal discussion with the participating individuals. The duration of each interview ranged between 30 and 40 min. Patients were categorized as having IBD (CD or UC) or as suffering from other benign chronic digestive diseases (irritable bowel syndrome or gastroesophageal reflux disease).

This study was conducted in accordance with the principles of the Declaration of Helsinki. The patients and control subjects were informed that participation was voluntary and that they could withdraw from the study at any time without consequence. This study was approved by our hospital’s ethics committee.

For statistical analysis, the results were expressed as the mean ± 1SD. The significance of the differences was calculated either with the Pearson chi-squared test or Student’s two-tailed *t*-test. The results were further evaluated by multivariate analysis to investigate the effect on the results of various factors such as age, gender, concurrent conventional medication, and disease duration. Differences of *p* < 0.05 (2-tailed) were considered significant. Calculations were performed using the SPSS statistical package (version 21.0).

## 3. Results

The results were analyzed by comparing the patients with IBD with the control group. Furthermore, patients with UC and patients with CD were also compared.

### 3.1. IBD Versus Disease Control Group

#### 3.1.1. Clinicoepidemiological Data

Table 1 shows the clinical and epidemiological data of the group of patients with IBD and the disease control group. As can be seen, there were more older women included in the disease control group compared to the IBD group. No significant differences were observed regarding many of the parameters such as educational level, smoking habit, history of allergy, use of contraceptives, marriage, breastfeeding, history of vitiligo and psoriasis, appendectomy, tonsillectomy, and presence of stress. Moreover, no significant differences were observed concerning the regular use of nutritional supplements and mastic gum and aloe vera (Table 1).

Significant differences were observed concerning the use of NSAIDs (9.6% vs. 24.6%, *p* = 0.004), the age of gestation both before and after the onset of the disease (*p* = 0.003), and the assumption of the positive role of stress in the recurrences of the underlying bowel situation (*p* = 0.015) (Table 1).

#### 3.1.2. Use of CAM

Table 2 shows the percentages of the IBD patients and those in the disease control group who used CAM. The rate of patients with IBD who used CAM at some point in their life was 36.3%, which although numerically greater than the disease control group (27.5%) did not significantly differ. Particularly for female patients, the differences in the most evident and relevant confounding variable, even though numerically more women with IBD used CAM compared to the disease control group, did not reach statistical significance (Table 2).

Regarding the individual components of CAM, significant differences between the two groups were observed regarding the use of chiropractic (*p* = 0.010), physical exercise (*p* = 0.003), massage (*p* = 0.013), and prayer (*p* = 0.014). Numerically, more of the control group used acupuncture, probiotics, vitamins, and aloe vera compared to IBD patients, although the differences did not reach statistical significance (Table 2).

Regarding the reasons that prompted the patients to use CAM, safety related to CAM therapy was significantly greater in the disease control group compared to the IBD patients (59.1% vs. 25.9%, *p* = 0.005). No other significant differences in the other reasons were noticed (i.e., desire for a holistic approach to disease, lack of response or unwanted effects of conventional treatment, and desire for better disease control). However, the side effects of using medicines and their effectiveness were greater in IBD patients compared to the disease control group, although differences were not statistically significant (26.5% vs. 13.6%, *p* = 0.268 and 25.0% vs. 19.0%, *p* = 0.776, respectively).

### 3.2. Crohn’s Disease Versus Ulcerative Colitis

#### 3.2.1. Clinicoepidemiological Data

Table 3 shows the clinical and epidemiological data of patients with CD and those with UC. As can be deduced from this table, the number of patients included in this study and their age did not differ significantly between the two groups. No significant differences were also observed regarding many of the other parameters, including educational level, smoking habit, history of allergy, use of contraceptives, marriage, breastfeeding, history of vitiligo and psoriasis, tonsillectomy, and the causative role of stress. However, as expected, significant differences were observed concerning the rate of appendectomy (28.4% vs. 5.9%, *p* = 0.001), the use of mesalazine (10.4% vs. 54.4%, *p* = 0.00001), and the use of biologic agents (46.3% vs. 23.5%, *p* = 0.007). Concerning smoking habits, although the percentage of smokers in CD was numerically higher compared to UC (25.4% vs. 11.8%), the differences did not reach statistical significance (Table 3).

#### 3.2.2. Use of CAM in Patients with CD and UC

Table 4 shows the percentages of the CAM use in patients with IBD. The percentage of patients with CD who used CAM at some point in their life was 34.3%, which although numerically smaller than the UC group (38.2%) was not significantly different.

Regarding the individual components of CAM, no differences between the patients with UC and CD were observed regarding all the factors studied, including the use of massage, reflexology, acupuncture, homeopathy, chiropractic, exercise, and prayer. No significant differences were observed regarding the use of probiotics and vitamins, nutritional supplements, and mastic gum and aloe vera (Table 4).

Regarding the reasons that prompted the patients with either CD or UC to use CAM, no significant differences were observed in all the factors studied.

In the multivariate analysis, the only parameters remaining significant regarding the use of CAM were university education (only in the use of CAM in general), female gender and treatment with biological agents (only in the use of special diets), homeopathic treatment (only university education), age (only exercise), and age (only in the use of probiotics) (Table 5). However, the multivariate analysis did not find the female gender to be a statistically significant variable of CAM use (95% CI 0.648–3.813. *p* = 0.317).

## 4. Discussion

The results of the present study revealed that a relatively high percentage of the patients with IBD (36.3%) had used some kind of CAM during the course of the disease. This proportion was numerically greater although not statistically significant than the CAM use from the disease control patients. Regarding the gender of the patients included in both groups, more female patients were included in the disease control group compared to the IBD group. This is probably due to two factors: the much higher proportion of female irritable bowel syndrome patients compared to men and the much higher willingness of female patients to participate in this study compared to men.

The most commonly used type of CAM was exercise (22.2%), followed by homeopathy (14.8%). The most widely used natural products by patients with IBD were probiotics, vitamins, and aloe vera. Those in the disease control group used chiropractic to a significantly greater extent than IBD patients. Also, numerically, more of the control group used acupuncture and massage than IBD patients. No significant differences were observed regarding the use of probiotics and vitamins.

This study also showed that 34.3% of CD patients used CAM, which although numerically smaller than the UC group (38.2%) was not significantly different. Regarding the reasons that prompted the patients to use CAM, significant differences between the two groups were observed only with regard to the safety factor.

Regarding the individual components of CAM in patients with either CD or UC, no differences were observed in all the factors studied, including the use of massage, reflexology, acupuncture, homeopathy, chiropractic, exercise, and prayer. No significant differences were also observed regarding the use of probiotics and vitamins, nor in the use of nutritional supplements and of mastic gum and aloe vera.

In the multivariate analysis, the parameters remaining significant were university education (in the use of CAM in general), sex and treatment with biologic agents (in the use of special diets), homeopathy (university education), and age (regarding exercise and in the use of probiotics).

Our results have similarities and differences from the results described in other countries. In the European Union, almost 26% of the population use CAM practices in general. Higher percentages were noticed in Germany (39.5%), Switzerland (39.4%), Austria (35.5%), and Finland (35.3%). The most widely used types of CAM were homeopathy, herbals, acupuncture, and reflexology. The most significant reasons for adopting this kind of treatment were skin disorders (38.1%), pain (38.0%), allergy (36.7%), digestive disorders (35.7%), headaches (34.1%), diabetes mellitus (23.6%), and depression (30.0%).

Concerning the use of CAM by patients with IBD in various countries of the world, the available data could be summarized as follows.

### 4.1. Northern Europe

The proportion of CAM use in countries of Northern Europe is almost uniform, fluctuating between 45 to 50%. The authors in a study from Germany calculated the use of CAM by IBD patients in two time periods (2002 and 2019). They found that, in 2019, 54% reported never using CAM, while 75% planned to use CAM in the future. A decrease in exclusive CAM use was found from 2002 (28%) to 2019 (16%). In logistic regression analysis, UC compared to CD; the side effects of conventional treatment, corticosteroids, or biologic agents; and low quality of life were significantly associated with CAM use in 2019 [8,9].

Similar results were obtained from Austria. The authors noticed that the prevalence of previous and current CAM use was 50.7%, and the results were not significantly different between CD and UC patients. In the multivariate analysis, the female gender and university education were independent indicators of CAM use. In contrast, the IBD-related indicators were longer disease duration and previous and current treatment with steroids and TNF-α inhibitors [7,8].

In a study from Hungary, the most frequently used forms of CAM were herbs (47.3%), homeopathy (14.6%), diet (12.2%), and acupuncture (5.8%). CAM use was associated in both UC and CD with a younger age, higher educational level, and use of immunosuppressants. In addition, the CAM use in UC was more common in women [9,10].

In a multicenter, controlled study from Sweden, 48.3% of IBD patients had used some type of CAM during the past year compared to 53.5% of the control group. The most frequently used type of CAM was massage (21.3% vs. 31.4% of the control group) and natural products (18.7% vs. 22.3% of the control group). Therefore, a significant percentage (48.3%) of Swedish patients with IBD used some CAM, although it was lower compared to the control group [10,11].

In a multicenter study from Norway, the authors noticed that 49% of IBD patients used some form of CAM in the last 12 months. Twenty-seven percent used services that offered CAM, while 21% adopted CAM personally without the help of a specialist. In this study, a significantly higher proportion of UC patients used CAM than CD patients (56% vs. 44%, respectively). Only co-morbidities appeared to be associated with the role of independent predictors of CAM use in UC patients. In patients with CD, however, the female sex, high level of education, and an age between 35 and 50 years were independent factors of CAM use [11,12]. Opheim et al., again in Norway, in a ten-year follow-up of patients with IBD, found that 30% reported using CAM at some time since their disease diagnosis, while 7.5% reported current CAM use. They also found that the proportion of CD patients who used CAM was greater than that of UC patients. Again, in this study, a young age, the female gender, and a high level of education were significant predictors of CAM use in UC patients, while a young age was the only predictor of CAM use in CD [12,13]. In a subsequent study, the same group of investigators found that the health-related quality of life in CAM users with IBD was significantly lower compared to non-users [13,14].

### 4.2. Southern Europe

Concerning the use of CAM in southern European countries, fewer patients use CAM than in northern Europe. A study from Italy found that 28% of patients with IBD reported using CAM, mainly homeopathy (43.6%), specific diets or nutritional supplements (35.5%), herbs (28.2%), exercise (25.6%), and prayer (14.7%). CAM was used to improve bowel symptoms (52.5%), hope for a full recovery (41%), and a reduction in medication intake (39.7%). The main reasons for CAM use were frequent recurrences and low communication with the responsible gastroenterologist [14,15].

Fernandez et al. in Spain found that 23% of patients with IBD had used CAM at some stage of the disease. The most common types of treatment involved herbal medicines, homeopathy, acupuncture, and kefir and aloe vera. Factors associated with CAM use were the presence of extraintestinal manifestations and long-term disease. Of interest was the finding that most patients felt that CAM did not substantially improve their health and that 11% of patients discontinued using conventional therapy [15,16].

In a study from France, the use of CAM practices was found to be relatively high (77.6%). The individual components of CAM were related to diet (30.7%), exercise (25.1%), homeopathic or traditional medicine (19.6%), naturopathy (15.2%), and mind–body medicine (9.1%). CAM users were more likely to have UC, be in clinical remission, have a high level of education, or have completed conventional therapy. Improvement in symptoms and quality of life was reported with all types of CAM [16,17].

### 4.3. North America

Investigators from Canada published three relevant studies concerning the countries of North America. In the first one, the Manitoba IBD Cohort Study, it was found that 74% of respondents used at least one CAM service or product during the 4.5-year follow-up period. Forty percent used CAM at some point, and 14% used CAM continuously. Female patients used CAM more often than males. There was no difference in the CAM use between patients with CD and those with UC. The most frequently used CAM was massage (30%), chiropractic (14%), physiotherapy (4%), acupuncture (3.5%), and homeopathy (3.5%). The most commonly used CAM products were probiotics (8%), oils (5.5%), glucosamine (4%), and chamomile (3.5%).

However, only 18% of consumers used CAM for IBD, so the majority chose it for other reasons. IBD patients usually try CAM, although very few have used it regularly [17,18]. In the second one, Li et al. found that, in patients with CD or UC, CAM use was associated with severe disease, use of CAM for other reasons, use of exercise and prayer for IBD, and a desire for an active role in decision treatment. CAM use was significantly higher at younger ages, in CD patients, and in UC patients who had a low degree of trust in their treating physician. The most common reasons for using CAM were the desire to control the disease, as well as a willingness to treat the condition holistically. The most common reasons for not using CAM were that conventional treatments were successful, people lacked knowledge about CAM, and the belief that CAM would not help. Thus, in Canadian IBD patients, disease activity and overall health-related attitudes and behaviors were associated with CAM use rather than demographic characteristics [18,19]. Finally, in the third study considered, Weizman et al. found that the prevalence of CAM use in IBD patients was 56% and did not differ significantly between CD and UC patients. The most common reason that prompted patients to use CAM was the ineffectiveness of conventional treatment (40%). The most frequently used type of CAM was probiotics (53%). CAM users had a higher educational level than non-users (75% vs. 62%). The CAM use in Canada was more frequent among those who experienced adverse effects with the use of conventional drugs, but this was without a reduction in the overall adherence to conventional therapy [19,20].

A study from the USA found that gastroenterologists with a special interest in IBD were receptive to patients’ use of CAM. The majority considered that CAM plays an essential role in the treatment of IBD, but this occurred without neglecting conventional treatment. Official initiatives with an educational nature, as well as social recommendations for the use of CAM, contribute significantly to adopting the correct use of CAM in daily practice. Of interest was the fact that the majority of respondents (65%) reported having attended formal educational events regarding the use of CAM. The majority also recommended probiotics, while almost half recommended acupuncture. Most respondents thought there is a complementary role for CAM use by IBD patients [20,21].

### 4.4. South America

The situation in countries of South America is entirely different. In a study from Brazil, the percentage of patients who used CAM was only 12.8%. No significant differences were found in the rates of CAM use between UC and CD patients. Interestingly, the authors included the consumption of tea, omega-3 fatty acids, and glutamine in the list of therapies that fell under the CAM category. If the use of these substances were not considered in evaluating the data, the proportion of CAM use would undoubtedly be considerably lower. The short time since diagnosis of the underlying IBD and poor quality of life were the independent factors associated with CAM use. This South American country had the lowest rate of CAM use among countries with comparable data [21,22].

In a study from Mexico, the rate of CAM use was 38.5%, of which 27% was used as an adjunct to conventional treatment. Acupuncture (42.8%), herbal products (35.7%), and homeopathy (35.7%) were the most frequently used methods. About 50% of patients used more than one method. The main reasons for using CAM were its use as an adjunct to conventional treatment (58.3%) and the absence of a significant degree of improvement with traditional therapy (33.3%). In 25% of cases, CAM was used after medical advice. No significant differences between CAM and non-CAM groups were found in the sociodemographic variables and clinical outcomes [22,23].

A study from Chile found that 25% of patients reported current CAM use, 30% reported past use, and 45% reported never having used CAM in the past. Forty-nine percent of patients informed the gastroenterologist that they used CAM. Notably, 86% of patients did not change their conventional treatment [23,24].

### 4.5. Asia

Thus far, scarce data regarding the use of CAM in Asia are available. Kim et al. noticed that 8.6% of IBD patients in East Asia (Japan, South Korea, and China) used CAM. A proportion of 29.7% of them used at least one type of CAM. As the authors stated, these percentages did not differ from the corresponding published ones concerning West Asian countries [24,25].

The available data from most studies refer to a specific period and do not answer whether the proportion of patients using CAM has changed over the years. Little data exist regarding the long-term use of CAM in patients with IBD, and the study of Lee et al. in Korea answered this question. These authors longitudinally followed 221 patients with IBD (2006 and 2014). They found that CAM use increased from 60.2% in 2006 to 79.6% in 2014, which was significant. A high level of education, previous failure of conventional treatment, and the use of corticosteroids were independent factors of CAM use [4]. In a second study from Korea, Park et al. found that 29.5% of patients with IBD reported CAM use, and 70.5% reported no CAM use after diagnosis of IBD. Using logistic regression analysis, university education, higher income levels, and longer duration of IBD were found to be independent predictors of CAM use. Among CAM users, only 28.7% discussed CAM use with their physician. Furthermore, 13.9% of CAM users discontinued conventional IBD therapy while using CAM [25,26].

### 4.6. Other Countries

A few other data sources from different countries are available. In a recent study from Saudi Arabia, the percentage of patients with IBD who used CAM was significantly more significant (54%) compared to that of the disease control group (43%). Additionally, the authors found that adherence to conventional medicine treatment recommendations was significantly lower in patients who used CAM compared to those who did not use CAM (35% vs. 11%) [26,27]. Another small study from Saudi Arabia found that 90% of this series reported using some form of CAM, of which 78% had used it within the previous year. Of note, 63% of patients reported taking CAM therapy without the knowledge of their treating physician. The most common source of advice on using CAM was relatives (66%), and the most common forms of CAM used were honey (62%), Zamzam water (54%), and physical exercise (32%). Binary logistic regression analysis found that diarrhea and taking azathioprine were independent predictors of CAM use [27,28].

Finally, Koning et al. in New Zealand found that 44.1% of IBD patients and 42.3% of control subjects used CAM during the previous year. The types of CAM most frequently used were vitamins (CD 25.2%, UC 23.7%, and control group 24.9%), followed by herbs (CD 15.1%, UC 15.2%, and control group 12.8%), and dietary supplements (CD, 8.5%, UC 12.6%, and control group 12.1%). Female gender, younger age, higher education, and higher income were independent predictors of CAM use in IBD patients. CAM use is common in New Zealand and does not differ between IBD patients and the general population. Socio-demographic factors, rather than disease phenotype, were found to be associated with CAM use in patients with IBD [28,29].

There are also two systematic reviews that were conducted dealing with the use of CAM in IBD patients. In the first one, published in 2015, Langhorst et al. included 26 randomized clinical trials and three clinical trials involving herbal therapy (aloe vera, Andrographis paniculata, artemisia absinthium, boswellia serrata, hemp, curcumin, evening primrose oil, Myrrhinil intest^®^, Plantago ovata, sophoramarintor, silymarin tor, wheatgrass juice, and wormwood); one randomized clinical trial with trichuris suis ovata; seven randomized clinical trials involving lifestyle modification, hypnotherapy, relaxation, and mindfulness techniques; and two randomized clinical trials involving the use of acupuncture. The results were satisfactory with herbs (Plantago ovata and curcumin in UC patients, wormwood in CD, and acupuncture in UC and CD) [29,30]. In the second one, Ng et al. recently investigated the quantity and assessed the quality of CAM clinical practice guidelines for IBD using the Appraisal of Guidelines for Research and Evaluation II instrument. Nineteen CAM clinical practice guideline recommendations for IBD were included. They found that most clinical practice guidelines were of low quality, and this was particularly true regarding the case for guidelines on the use of CAM in IBD [30,31].

The general conclusion that could be drawn from the data mentioned above is that although the proportion of patients with IBD using CAM differs significantly in various countries (e.g., there is a “gap” between countries of North and South Europe and countries of North and South America), the factors driving IBD patients to adopt the use of CAM are essentially similar. As was also noticed in our study, the main reasons concern the high level of education and younger ages. The reasons for adopting CAM in our study were quite similar to other studies regarding the safety of CAM and the possibility of side effects of conventional treatment being among the most important. It should be stressed that some CAM therapies are supported by the data from randomized controlled clinical trials and the data from meta-analyses and systematic reviews [29,30]. Also, a review of the most significant clinical trials of herbal therapies for patients with IBD described the positive results obtained via the administration of aloe vera, polyphenols (green tea), wheatgrass, bilberry, Boswellia serrata, and cannabis [31,32]. However, we must bear in mind that some CAM therapies based on the administration of herbs can cause unwanted side effects, as well as interaction with conventional therapy [32,33], and that CAM use may indicate psychosocial impairment in patients with IBD [33,34]. The psychophysiological vulnerability of IBD patients, possibly due to the presence of mood disorders, anxiety, increased stress, or maladaptive coping strategies, underscores the psychological needs of IBD patients [34,35]. Another primary concern related to CAM practices is that patients’ disclosure of CAM use to their physicians remains low. This could have serious consequences as 13.9% of IBD patients avoid using conventional therapy alongside CAM. A certain number of IBD patients did not consider CAM as a natural treatment, thus avoiding mentioning it. Sometimes, the gastroenterologist does not consider therapies such as diet, lifestyle, biofeedback, hypnosis, relaxation techniques, etc., as components of CAM treatment [35,36]. Another point of critical significance is that CAM subjects are not widely taught in medical schools, nor are they available in the vast majority of hospitals [36,37], even though their use has increased worldwide and is still increasing. For example, in the United States, it has been estimated that almost 59 million adults and children have used at least one form of CAM, according to the U.S. National Center for Health Statistics (NHSR 95/2016). In many countries, the lack of legislation regarding licensing, registration sales, insurance coverage, safety and efficacy, post-market surveillance, quality assurance, and the lack of funds for research were the main barriers to the lack of data [37,38].

The available data in the international literature on the attitude of gastroenterologists following-up IBD patients toward CAM are scarce. An earlier study from the UK assessed the responses of 2748 individuals, of whom 79% practiced in the NHS, 32% practiced CAM themselves, and 41% referred patients to CAM. CAM was found to be used by private practitioners rather than by NHS doctors. Acupuncture, aromatherapy, and chiropractic were the most commonly used treatments. Attitudes toward CAM were generally positive, particularly among those involved in palliative care, rehabilitation, nuclear medicine, and urogenital medicine. The authors concluded that at least one in ten UK specialist doctors were actively involved in CAM therapies for this period, although only 13% had received any CAM training [39]. In a study of gastroenterologists’ attitudes toward CAM, Gallinger ZR et al. found that nearly a third of respondents reported discussing CAM with their patients, but 90% felt that most patients were reluctant to discuss it. A proportion of 72% felt comfortable discussing CAM, while those who did not cited a lack of relevant knowledge as the most common reason. A proportion of 65% of gastroenterologists noted that they had no training in CAM, while 50% had previously recommended acupuncture. The vast majority felt that CAM can serve as adjunctive therapy in IBD. Academic IBD specialists were receptive to the use of CAM, and most believed it had a role in the treatment of IBD without necessarily compromising conventional treatment [21]. A multicenter study from Italy found that, of the 438 study participants, 42% believed CAM could have a unifying role in the context of CM. The most knowledgeable physicians about CAM were oncologists. Even physicians in research institutes or university hospitals were more informed than their general hospital colleagues. In addition, 55% of study participants recommended CAM interventions to their patients, while 44% discussed CAM implementation with them [40].

Concerning the situation in Greece, scarce data on the attitude of Greek physicians toward CAM are available. It seems that Greek doctors show little or no interest in alternative therapies but are familiar to some degree with homeopathy (59%), special diets (58%), and acupuncture (48%) [38,41]. The perceived rates of patients using CAM are also low. However, it seems that, in the future, the media and the Internet may generate a greater demand for these methods in Greece, and physicians will thus need to be better informed. In the thesis of D. Stavropoulou (2015) entitled “Methods of Alternative Medicine: Practical Applications in Greece”, it is stated that “…In essence, in Greece, holistic medicine is not funded; rather, the state tries to provide financial assistance to its users. Additionally, application of alternative medicine is barely advanced; its adherents receive it mainly through private practitioners, for example, visits to the premises of a psychotherapist, acupuncturist, home therapist, etc. The costs of such visits or sessions (if treatment is planned and implemented) amount to 50 to 60 Euros per session…”.

Concerning the drawbacks of this study, we considered the differences in the age between the IBD group and disease control group as significant but relatively unavoidable. In future studies, attention should be paid to the patterns of CAM use in IBD versus the control group within each homogeneous subgroup of the cohorts (e.g., sex, age, educational level, etc.).

## 5. Conclusions

The conclusion that emerges from the results of our study and the relevant international literature is that the use of CAM is more widespread among young people and people with a high level of education. However, some design weaknesses of the published studies, and notably the absence of an economic evaluation of CAM, result in difficulties in drawing firm and valuable conclusions. Other facts concern the lack of relevant legislative regulations, the control and assessment of the various herbal preparations by individual health authorities, and the granting of marketing authorizations. The duty of the health authorities, apart from offering notification of marketing, is to evaluate the effectiveness and safety of circulating preparations, post-marketing surveillance, and quality assurance [37,38]. At the same time, pharmaceutical companies should financially support international multicenter studies to adequately investigate the effectiveness of CAM treatment either as a sole or as a parallel treatment with the conventional ones.

## Figures and Tables

**Table 1 nutrients-16-03679-t001:** The clinical and epidemiological parameters of patients with IBD and those in the disease control group.

Parameter	IBD	Disease Control	*p*-Value
Number	270	138	
Sex Men women	142 (77.2) 128 (24.8)	42 (30.4) 96 (69.6)	*p* = 0.003
Age (mean ± 1SD)	42.3 ± 17.5	52.9 ± 16.7	*p* = 0.003
Marriage Yes Widow or widower Divorce	142 (52.6) 6 (2.2) 10 (3.7)	90 (65.2) 6 (4.3) 6 (4.3)	0.171
Education University High Moderate Elemental	124 (45.9) 44 (16.3) 66 (24.4) 36 (13.3)	64 (46.4) 30 (21.7) 24 (17.4) 20 (14.5)	0.610
Smoking Smoker Never Ex-smoker	50 (18.5) 136 (50.4) 84 (31.1)	24 (12.4) 84 (60.9) 30 (21.7)	0.301
Use of contraceptives	44 (34.4)	30 (31.3)	0.840
Breastfeeding	214 (79.3)	112 (81.2)	0.854
Use of NSAIDs	26 (9.6)	34 (24.6)	*p* = 0.004
History of allergy	104 (38.5)	62 (44.9)	0.378
Vitiligo	2 (0.7)	2 (1.4)	1.000
Psoriasis	14 (5.2)	2 (1.4)	0.270
Causative role of stress	226 (84.3)	68 (73.9)	0.126
Role of stress on recurrences	224 (88.2)	60 (71.4)	*p* = 0.015
Gestation before the onset of the disease	56 (45.2)	60 (71.4)	*p* = 0.046
Gestation after the onset of the disease	24 (20.7)	0 (0.0)	*p* = 0.015
Appendectomy	46 (17.0)	30 (21.7)	0.450
Tonsillectomy	58 (21.5)	30 (21.7)	1.000

(Numbers in parentheses are percentages).

**Table 2 nutrients-16-03679-t002:** Use of CAM in the patients and those in the disease control group.

Parameter	IBD	Disease Control	*p*-Value
Use of CAM Use of CAM (women)	98 (36.3) 52 (40.6)	38 (27.5) 32 (33.3)	0.272 0.554
Prayer	38 (14.1)	4 (2.9)	*p* = 0.014
Chiropractic	2 (0.7)	16 (11.6)	*p* = 0.001
Massage	6 (2.2)	14 (10.1)	*p* = 0.013
Acupuncture	14 (5.2)	16 (11.6)	0.097
Homeopathy	40 (14.8)	16 (11.6)	0.668
Reflexology	10 (3.7)	4 (2.9)	1.000
Exercise	60 (22.2)	8 (5.8)	*p* = 0.003
Probiotic consumption	146 (54.1)	56 (40.6)	0.068
Vitamin consumption	152 (56.3)	96 (69.6)	0.066
Use of nutritional supplements	68 (25.2)	32 (23.2)	0.864
Use of mastic gum or aloe vera	80 (29.6)	30 (21.7)	0.248

**Table 3 nutrients-16-03679-t003:** Clinical and epidemiological parameters of the patients with CD and UC.

Parameter	Crohn’s Disease	Ulcerative Colitis	*p*-Value
Number	134	136	
Sex Men Women	68 (50.7) 66 (49.3)	74 (54.4) 62 (45.6)	0.731
Age (±1SD)	41.6 +/− 17.9	43.0 ± 17.1	0.731
Marriage Yes Widow or widower Divorce	66 (49.3) 4 (3.0) 2 (1.5)	76(55.9) 2 (1.5) 8 (5.9)	0.373
Education University High Moderate Elemental	58 (43.3) 26 (19.4) 36 (26.9) 14 (10.4)	66 (48.5) 18 (13.2) 30 (22.1) 22 (16.2)	0.544
Positive family history for IBD	20 (14.9)	12 (8.8)	0.300
Smoking Smoker Never Ex-smoker	34 (25.4) 62 (46.3) 38 (28.4)	16 (11.8) 74 (54.4) 46 (33.8)	0.126
Use of contraceptives	18 (27.3)	26 (41.9)	0.294
Breast feeding	102 (76.1)	112 (82.4)	0.403
Use of NSAIDs	16 (11.9)	10 (7.4)	0.398
History of allergy	54 (40.3)	50 (36.8)	0.725
Vitiligo	2 (1.5)	0 (0.0)	0.496)
Psoriasis	8 (6.0)	6 (4.4)	0.718
Causative role of stress	110 (82.1)	116 (86.6)	0.635
Role of stress on recurrences	110 (82.1)	114 (86.4)	0.589
Gestation before the onset of the disease	32 (50.0)	24 (40.0)	0.456
Gestation after the onset of the disease	10 (16.7)	14 (25.0)	0.525
Appendectomy	38 (28.4)	8 (5.9)	*p* = 0.001
Tonsillectomy	36 (26.9)	22 (16.2)	0.147
Nutritional support	40 (29.9)	28 (20.6)	0.239
Use of mastic gum and/or aloe vera	42 (31.3)	38 (27.9)	0.709
Treatment Biologic agents Immunosuppressives Mesalazine Corticosteroids No treatment	62 (46.3) 22 (16.4) 14 (10.4) 8 (6.0) 28 (22.4)	32 (23.5) 18 (13.2) 74 (54.4) 12 (8.8) 10 (7.4)	*p* = 0.007 0.636 *p* = 0.00001 0.744 0.016
Active disease	56 (41.8)	48 (35.3)	0.482

(Numbers in parentheses are percentages).

**Table 4 nutrients-16-03679-t004:** Use of CAM in patients with Crohn’s disease and ulcerative colitis.

Parameter	Crohn’s Disease	Ulcerative Colitis	*p*-Value
Number	134	136	
Use of CAM (one or more)	46 (34.3)	52 (38.2)	0.721
Prayer	18 (13.4)	20 (14.7)	1.000
Chiropractic	0 (0.0)	2 (1.5)	1.000
Massage	4 (3.0)	2 (1.5)	0.619
Acupuncture	6 (4.5)	8 (5.9)	1.000
Homeopathy	20 (14.9)	20 (14.7)	1.000
Reflexology	6 (4.5)	4 (2.9)	0.680
Exercise	26 (19.4)	34 (25.0)	0.535
Probiotic consumption	68 (50.7)	78 (57.4)	0.492
Use of vitamins	76 (56.7)	76 (55.9)	1.000
Reasons for use of CAM Safety Side-effects of medicines Effectiveness Manipulation of treatment Interruption of corticosteroids	24 (30.0) 24 (31.6) 18 (23.1) 16 (20.5) 16 (20.0)	20 (22.2) 20 (22.2) 24 (26.7) 20 (22.2) 12 (13.3)	0.464 0.455 0.803 1.000 0.560

(Numbers in parentheses are percentages).

**Table 5 nutrients-16-03679-t005:** Results of the multivariate analysis.

	Variable	*p*-Value	95% CI Lower Upper
Use of CAM (general)	University education	0.009	1.936–107.984
Use of special diet	Female gender Treatment with biologic agents	0.029 0.044	1.101–5.869 1.037–13.848
Homeopathic	University education	0.046	1.050–215.700
Exercise	Age	0.004	0.866–0.973
Probiotic consumption	Age	0.028	0.939–0.996
Vitamin consumption	Treatment with CS	0.045	1.055–109.459

## Data Availability

The study data are unavailable due to privacy issues.

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
