# Peer review of "Use of Complementary and Alternative Medicine by Greek Patients with Inflammatory Bowel Disease"

_nutrients, 2024, doi:10.3390/nu16213679_

Round 1
Reviewer 1 Report
Comments and Suggestions for Authors
This study addresses an important yet often overlooked issue in the management of IBD. It finds a remarkably high use of CAM among IBD patients in Greece, and it identifies a number of characteristics that appear to be associated with its use.
It is of paramount importance in any comparison between groups to ensure that the groups are comparable in the distribution of most potentially confounding factors. The most obvious disparity in the composition of the IBD and control cohorts in this study was the sex distribution; indeed, a major conclusion was that women (among other subsets) were the most likely users of CAM.
This observation was confirmed, the authors tell us, by “multivariable analysis.” Yet, in careful scrutiny of the manuscript, I had difficulty in discerning whether or not women per se demonstrated differences in CAM use between the IBD and control groups. It seems to me that the principal aim of this paper was to identify any differences in CAM utilization between IBD patients and controls; yet it was not clear to me, when looking exclusively at the women in both cohorts, whether or not this subsets manifested any such differences.
The authors understandably give considerable attention to the patterns of CAM utilization in different countries. This question is clearly relevant in view of the role of cultural influences in the use of CAM. Yet I feel that similar attention should be paid to the patterns of CAM use in IBD versus controls within each homogeneous subgroup of the cohorts (e.g., sex, age, educational level, etc.). If this consideration is already expounded in the authors’ discussion, and I have missed it, I apologize in advance!
Author Response
Reviewer 1
- “This study addresses an important yet often overlooked issue in the management of IBD. It finds a remarkably high use of CAM among IBD patients in Greece, and it identifies a number of characteristics that appear to be associated with its use. It is of paramount importance in any comparison between groups to ensure that the groups are comparable in the distribution of most potentially confounding factors. The most obvious disparity in the composition of the IBD and control cohorts in this study was the sex distribution; indeed, a major conclusion was that women (among other subsets) were the most likely users of CAM. This observation was confirmed, the authors tell us, by “multivariable analysis.” Yet, in careful scrutiny of the manuscript, I had difficulty in discerning whether or not women per se demonstrated differences in CAM use between the IBD and control groups. It seems to me that the principal aim of this paper was to identify any differences in CAM utilization between IBD patients and controls; yet it was not clear to me, when looking exclusively at the women in both cohorts, whether or not these subsets manifested any such differences”.
Answer:
About the gender of the patients included in both groups, more female patients were indeed included in the disease control group compared to the IBD group. In our opinion, this is due to two factors: the much higher proportion of female irritable bowel syndrome patients compared to men and the much higher willingness of female patients to participate in the study compared to men. The above two observations were added to the discussion section of the article. We thank the reviewer very much for this important observation regarding CAM use by female IBD patients. Indeed, only the use of specific diets was the parameter significantly associated in the multivariate analysis with female gender in IBD patients. This note was added to the article's discussion, and the relevant section in the article's abstract was corrected.
- “The authors understandably give considerable attention to the patterns of CAM utilization in different countries. This question is clearly relevant in view of the role of cultural influences in the use of CAM. Yet I feel that similar attention should be paid to the patterns of CAM use in IBD versus controls within each homogeneous subgroup of the cohorts (g., sex, age, educational level, etc.). If this consideration is already expounded in the authors’ discussion, and I have missed it, I apologize in advance!”
Answer:
We did not perform this comparison in homogeneous subgroups, and we did not find relevant data in the literature. However, we consider this observation of the utmost importance. The relevant paragraph describing our study's drawbacks, especially regarding this point, was added at the discussion part of the paper.

Reviewer 2 Report
Comments and Suggestions for Authors
The manuscript by John K. Triantafillidis investigated the use of complementary and alternative medicine by Greek patients with inflammatory bowel disease. The study is interesting and informative. I have the following questions and comments:
1, the statistics of the study must be further detailed.
2, what kind of CAM was included in the study? This must be specified.
3, did the authors detect any side effects or adverse effects about the use of CAM in patients with inflammatory bowel disease?
4, what are the doctors attitude towards the use of CAM in patients with inflammatory bowel disease? This should be discussed.
5, why patients with inflammatory bowel disease use CAM in Greek?
Author Response
Reviewer 2
The manuscript by John K. Triantafillidis investigated the use of complementary and alternative medicine by Greek patients with inflammatory bowel disease. The study is interesting and informative. I have the following questions and comments:
- The statistics of the study must be further detailed.
Answer:
The statistics were re-examined, and any oversights, even minor ones, were re-examined.
- What kind of CAM was included in the study? This must be specified.
Answer:
In the Materials and Methods section it is stated that “…The questionnaire included 30 questions about the type and frequency of CAM use. We recorded queries related to the use of plants and herbs (e.g., mastic gum, aloe), lifestyle modifications (e.g., exercise, diet, smoking), as well as psychosomatic therapies, massage, kinesiology, homeopathy, and acupuncture, and data concerning demographic parameters (e.g. age, gender, education, marital status, employment status, diet, and lifestyle habits). The questionnaire also included disease characteristics, type of IBD, current activity, duration of disease, and type of conventional medications. One of the authors (JKT) completed the questionnaire in all patients and controls, following a detailed personal discussion with the participating individuals. The duration of each interview ranged between 30 and 40 minutes…”
The following paragraph was added in the Introduction part: “According to the U.S. National Center for Complementary and Integrative Health (NCCIH), Complementary and Alternative Medicine (CAM) therapies include a wide spectrum of practices and products, either biological (e.g., herbs or botanicals, vitamins, minerals, probiotics, homeopathic products, and Chinese herbal remedies) or non-biological (e.g., prayer, meditation, music therapy, yoga)”.
- Did the authors detect any side effects or adverse effects about the use of CAM in patients with inflammatory bowel disease?
Answer:
Unfortunately, the questionnaire did not include the type of side effects from CAM use, so the question cannot be answered.
- What are the doctors’ attitudes towards the use of CAM in patients with inflammatory bowel disease? This should be discussed.
Answer:
The relevant data of the literature was added (in the discussion part).
- Why patients with inflammatory bowel disease use CAM in Greece?
Answer:
Patients with IBD used CAM because they were relatively certain about the high level of safety of CAM treatments, the high incidence of side effects with conventional treatment, the possibility of achieving better results with CAM, and the desire to manage their own disease treatment.
Concerning the relevant percentages in the results section it is stated that “…Regarding reasons for using CAM, safety related to CAM therapy was statistically significantly greater in the disease control group compared to IBD patients (59.1% vs. 25.9%, P=0.005).

Round 2
Reviewer 1 Report
Comments and Suggestions for Authors
“We did not perform this comparison in homogeneous subgroups.” Why not? All that is required is to compare CAM use between female IBD patients and female controls, thus correcting for the most obvious and relevant confounding variable.
Author Response
Below are our responses to the comments and reviewers' observations (Round 2). Again we deeply appreciate the reviewers' diligent efforts in evaluating our article.
Round 2 review report from Reviewer 1
- “We did not perform this comparison in homogeneous subgroups.” Why not? All that is required is to compare CAM use between female IBD patients and female controls, thus correcting for the most obvious and relevant confounding variable.
Answer:
Statistical analysis was performed between women with IBD and women in the control group. No statistically significant differences were found (Table 2 of the text), and multivariate analysis also showed no statistically significant differences.
Round 2 review report from Reviewer 2
- The authors have revised the manuscript accordingly. It can be considered for publication. Just one more suggestion, the tables should be changed to a three-line format.
Answer:
Tables were changed to a three-line format

Reviewer 2 Report
Comments and Suggestions for Authors
The authors have revised the manuscript accordingly. It can be considered for publication. Just one more suggestion, the tables should be changed to a three-line format.
Author Response

(The authors gave the same response as above.)
